# Benefits of NGS in Advanced Lung Adenocarcinoma Vary by Populations and Timing of Examination

**DOI:** 10.3390/ijms25136949

**Published:** 2024-06-25

**Authors:** Po-Hsin Lee, Wei-Fan Ou, Yen-Hsiang Huang, Kuo-Hsuan Hsu, Jeng-Sen Tseng, Gee-Chen Chang, Tsung-Ying Yang

**Affiliations:** 1Division of Chest Medicine, Department of Internal Medicine, Taichung Veterans General Hospital, Taichung 407, Taiwan; berry7bo@gmail.com (P.-H.L.); spmanner@hotmail.com (W.-F.O.); waynehuang0622@gmail.com (Y.-H.H.); jonyin@gmail.com (T.-Y.Y.); 2School of Medicine, National Yang Ming Chiao Tung University, Taipei 112, Taiwan; 3Doctoral Program in Translational Medicine, National Chung Hsing University, Taichung 402, Taiwan; 4Rong Hsing Translational Medicine Research Center, National Chung Hsing University, Taichung 402, Taiwan; 5Institute of Biomedical Sciences, National Chung Hsing University, Taichung 402, Taiwan; 6Lung Cancer Comprehensive Care and Research Center, Taichung Veterans General Hospital, Taichung 407, Taiwan; vghryan@gmail.com; 7Division of Critical Care and Respiratory Therapy, Department of Internal Medicine, Taichung Veterans General Hospital, Taichung 407, Taiwan; 8Department of Post-Baccalaureate Medicine, College of Medicine, National Chung Hsing University, Taichung 402, Taiwan; 9Division of Pulmonary Medicine, Department of Internal Medicine, Chung Shan Medical University Hospital, Taichung 402, Taiwan; geechen@gmail.com; 10School of Medicine, Chung Shan Medical University, Taichung 402, Taiwan; 11Institute of Medicine, Chung Shan Medical University, Taichung 402, Taiwan; 12Department of Life Sciences, National Chung Hsing University, Taichung 402, Taiwan

**Keywords:** lung cancer, adenocarcinoma, next-generation sequencing (NGS), driver mutation, targeted therapy

## Abstract

Despite the widespread application of next-generation sequencing (NGS) in advanced lung adenocarcinoma, its impact on survival and the optimal timing for the examination remain uncertain. This cohort study included advanced lung adenocarcinoma patients who underwent NGS testing. We categorized patients into four groups: Group 1: treatment-naïve, upfront NGS; Group 2: Treatment-naïve, exclusionary *EGFR*/*ALK*/*ROS1*; Group 3: post-treatment, no known *EGFR*/*ALK*/*ROS1*; Group 4: known driver mutation and post-TKI treatment. A total of 424 patients were included. There were 128, 126, 90, and 80 patients in Groups 1, 2, 3, and 4, respectively. In Groups 1, 2, 3, and 4, targetable mutations were identified in 76.6%, 49.2%, 41.1%, and 33.3% of the patients, respectively (*p* < 0.001). Mutation-targeted treatments were applied in 68.0%, 15.1%, 27.8%, and 22.5% of the patients, respectively (*p* < 0.001). In the overall population, patients receiving mutation-targeted treatments exhibited significantly longer overall survival (OS) (aHR 0.54 [95% CI 0.37–0.79], *p* = 0.001). The most profound benefit was seen in the Group 1 patients (not reached vs. 40.4 months, *p* = 0.028). The median OS of patients with mutation-targeted treatments was also significantly longer among Group 2 patients. The median post-NGS survival of patients receiving mutation-targeted treatments was numerically longer in Group 3 and Group 4 patients. In conclusion, mutation-targeted therapy is associated with a favorable outcome. However, the opportunities of NGS-directed treatment and the survival benefits of mutation-targeted treatment were various among different populations.

## 1. Introduction

Lung cancer is the leading cause of cancer-related death worldwide [1]. The treatment of lung cancer has been directed toward personalized therapy. Advances in lung cancer treatment have led to improvements in patient outcomes [2]. In addition to assessments of tumor stage and performance status, treatment decisions are largely based on pathological classification, immunological biomarkers, as well as the results of genetic analysis [3,4]. Adenocarcinoma is the most common histological type of lung cancer, and increasing knowledge of genetic profiling has led to the emergence of targeted therapy as a powerful treatment strategy. For advanced treatment-naïve lung adenocarcinoma patients, driver-targeted therapy has been proven to improve the outcome [5].

The comprehensive molecular profiling of genetic alterations plays a crucial role in the selection of corresponding targeted therapy and has been suggested to be a mandatory examination at the time of lung cancer diagnosis. Currently, at least nine driver mutations are deemed to be druggable, according to the clinical practice guidelines for non-small cell lung cancer (NSCLC) [3,4]. Various methods could be used to detect these genetic alterations. Next-generation sequencing (NGS) is a powerful technology for DNA and RNA sequencing and mutation detection. As compared with conventional methods, such as polymerase chain reaction (PCR)-based platforms, immunohistochemistry (IHC) staining, and fluorescence in situ hybridization (FISH), NGS possesses the following benefits: single input of DNA/RNA, quantitative, high sensitivity, high specificity, and comprehensive genomic coverage. However, NGS has some disadvantages; it is more expensive as well as time- and tumor tissue-consuming [6].

Apart from the front-line therapies, NGS could be applied for patients with progression on certain targeted therapies [7,8,9] in order to help identify the resistance mechanisms, predict the outcome, and guide the subsequent therapy. Although the clinical applications of NGS are growing, the affordability of testing and the availability of tumor specimens remain important issues in clinical practice [10]. More importantly, the survival benefits of NGS remain uncertain and may vary among different situations [11,12,13]. There are still controversies regarding who should undergo NGS testing and when these tests should be used. Herein, we evaluated the benefits of NGS in advanced lung adenocarcinoma patients, particularly focusing on its impact on various populations and the timing of examination.

## 2. Results

### 2.1. Patients and Their Demographic Data

The flowchart of patient selection, categorization, and outcome analysis is shown in Figure 1. During the period between December 2018 and January 2024, there was a total of 611 lung cancer patients with NGS results. Among them, 76 with duplicate data, 62 with non-adenocarcinoma histology, 32 with early-stage NSCLC receiving curative treatment, 10 with incomplete survival follow-up data, and 7 with active malignancy other than lung cancer were excluded. Finally, a total of 424 lung adenocarcinoma patients with available NGS results were included in the analysis. The baseline characteristics are summarized in Table 1. The median age was 61.6 years. In total, 208 patients (49.1%) were female, 241 patients (56.8%) were never smokers, while 408 patients (96.2%) had stage IV disease. At the diagnosis of advanced lung cancer, the ECOG PS was 0–1 in 371 patients (87.5%). In terms of NGS testing, tumor tissue and liquid biopsy were used in 211 (49.8%) and 213 (50.2%) patients, respectively. Group 1, Group 2, Group 3, and Group 4 had 128 (30.2%), 126 (29.7%), 90 (21.2%), and 80 (18.9%) patients, respectively.

### 2.2. Distribution of Targetable Mutation(s) among Various Populations

The distribution of targetable mutation(s) among the four groups of patients is summarized in Figure 2. In the Group 1 population, *EGFR* mutation accounted for the most common genetic alteration (54.7%), which was followed by *ALK* fusion (6.3%), *HER2* mutation (6.3%), and *ROS1* fusion (4.7%). Of these patients, 20.3% did not have detectable driver mutations and 3.1% of patients had untargetable mutations. Among patients in the Group 2 population, *HER2* mutation (13.5%) and *MET* aberration (13.5%) were the two most common targetable mutations, which is followed by *EGFR* mutation (9.5%), *KRAS* G12C mutation (6.3%), *BRAF* V600E mutation (3.2%), and *ROS1* fusion (2.4%). A total of 34.1% of Group 2 patients had no detectable driver mutations, and 16.7% of patients had untargetable mutations. 

*EGFR* mutation (21.1%) was the most common genetic alteration in Group 3 patients, which was followed by *HER2* mutation (7.8%), *ROS1* fusion (6.7%), *KRAS* G12C mutation (2.2%), *ALK* fusion (1.1%), *MET* aberration (1.1%), and *RET* fusion (1.1%). Overall, 13.3% of patients had untargetable mutations, and 45.6% of patients did not have detectable driver mutation. Among the Group 4 patients, T790M (13.6%) and *MET* aberration (11.1%) were the two most common targetable mutations. As high as 66.7% of the NGS results in Group 4 patients were deemed to be untargetable or undetectable.

### 2.3. Survival Benefits of Mutation-Targeted Treatment in Overall Population

The results of survival outcomes are shown in Appendix A and Figure 3. In the overall population, a history of mutation-targeted treatment was associated with a significantly longer OS (59.9 months [95% CI 49.4–70.4] vs. 43.1 months [95% CI 35.0–51.3], log-rank *p* = 0.002) (Appendix A). In the multivariate analysis adjusting for age, gender, smoking behavior, ECOG PS, tumor stage, and NGS sample types, mutation-targeted treatment remained an independent predictor of a favorable outcome (aHR 0.54 [95% CI 0.37–0.79], *p* = 0.001).

### 2.4. Various Opportunities of Receiving NGS-Directed Treatment among Different Populations

Because the clinical and genetic characteristics of the four groups of patients were not the same, we further analyzed the clinical benefits of mutation-targeted treatment among the four groups. The results of the percentage of targetable mutation(s) and mutation-targeted treatment among various patient populations are shown in Figure 4. The percentages of potential targetable mutation(s) were 76.6%, 49.2%, 41.1%, and 33.3% in the Group 1, Group 2, Group 3, and Group 4 patients, respectively (*p* < 0.001). The percentages of mutation-targeted treatment were 68.0%, 15.1%, 27.8%, and 22.5% in Group 1, Group 2, Group 3, and Group 4 patients, respectively (*p* < 0.001). Treatment-naïve patients with upfront NGS testing had the highest likelihood of having targetable mutation(s) and receiving the corresponding targeted therapies.

### 2.5. Various Survival Benefits of Mutation-Targeted Treatment among Different Populations

The results of survival outcomes among the four groups of patients are shown in Figure 3 and Table 2. The overall survival and post-NGS survival time of the treatment-naïve populations (Group 1 and Group 2) and treated populations (Group 3 and Group 4) were analyzed, respectively. Among Group 1 patients, mutation-targeted treatment was associated with significantly longer OS (not-reached [NR] months [95% CI NR-NR] vs. 40.4 months [95% CI NR-NR]), log-rank *p* = 0.028). In these patients, EGFR mutation treated with EGFR-TKI (*n* = 68, 78.2%) was the most common treatment modality, which was followed by *ALK* fusion/ALK inhibitors (*n* = 8, 9.2%) and *ROS1* fusion/ROS1 inhibitors (*n* = 6, 6.9%). In the subgroup analysis of patients with targetable mutation(s), the OS of subjects with corresponding targeted therapy was significantly longer than that of patients without targeted therapy (*p* < 0.001) (Appendix A).

Among Group 2 patients, mutation-targeted treatment was also associated with significantly longer OS (19.2 months [95% CI 13.6–24.6] vs. 12.4 months [95% CI 8.3–16.6], log-rank *p* = 0.049). In Group 2, *EGFR* mutation treated with EGFR-TKI (*n* = 8, 42.1%) was the most common treatment modality, which was followed by *MET* aberration and its inhibitor(s) (*n* = 6, 31.6%). Other mutation-targeted treatment included *HER2* mutation, *ROS1* fusion, *BRAF* V600E mutation, as well as *KRAS* G12C mutation. In the subgroup analysis of patients with targetable mutation(s), subjects with corresponding targeted therapy generally experienced a longer OS compared to those without targeted therapy (*p* = 0.037) (Appendix A).

Among Group 3 patients, *EGFR* mutation treated with EGFR-TKI (*n* = 16, 64.0%) was the most common mutation-targeted treatment, which was followed by *ROS1* fusion (*n* = 4, 16.0%) and *HER2* mutation (*n* = 3, 12.0%). Patients receiving mutation-targeted treatment experienced a numerically longer post-NGS survival of (49.9 months [95% CI 18.4–81.3] vs. 38.1 months [95% CI NR-NR], log-rank *p* = 0.684). In the subgroup analysis of patients with targetable mutation(s), the post-NGS survival of subjects with corresponding targeted therapy was numerically longer than that of patients without targeted therapy (49.9 months vs. 24.2 months) (Appendix A).

Among the Group 4 patients, second line osimertinib treatment for *EGFR* T790M-positive individuals (*n* = 10, 55.6%) was the most common treatment modality, which was followed by post-osimertinib *MET* amplification treated with MET inhibitor(s) (*n* = 7, 38.9%). Patients receiving mutation-targeted treatment experienced a numerically longer post-NGS survival (NR months [95% CI NR-NR] vs. 19.3 months [95% CI 16.6–22.1], log-rank *p* = 0.377). In the subgroup patients with progression on first- or second-generation EGFR-TKI, the post-NGS survival of subjects with positive T790M treated with osimertinib was numerically longer than that of other patients (NR vs. 35.2 months) (Appendix A). Among the subgroup patients with progression on osimertinib, the post-NGS survival of individuals with *MET* amplification treated with MET inhibitor(s) was also numerically longer (NR vs. 19.1 months) (Appendix A).

## 3. Discussion

Although NGS has been widely applied in the management of advanced lung adenocarcinoma, its impact on survival and the optimal timing for the examination remain uncertain. In this study, we categorized our patients into four groups according to the patients’ characteristics and the timing of NGS examination. We found that regardless of the timing of NGS examination, a high proportion of targetable mutations could be identified. The Lung Cancer Mutation Consortium conducted by Kris et al. prospectively enrolled advanced lung adenocarcinoma patients to detect ten driver mutations, and the results indicated that patients with detectable driver mutation and receiving a corresponding targeted therapy could experience a significantly longer survival time [5]. Herein, our results also revealed that mutation-targeted treatment independently predicted a favorable outcome in advanced lung adenocarcinoma. Moreover, we found that the percentage of actionable mutations, the opportunities of NGS-directed treatment, and the survival benefits of mutation-targeted treatment varied among the different patient populations.

In the present study, Group 1 contained treatment-naïve lung adenocarcinoma patients with upfront NGS testing; hence, the distribution of driver mutations was similar to those observed in previous studies [14,15]. The Group 1 population possessed the highest detection rate of targetable mutations and had the greatest opportunity of receiving mutation-targeted treatment. Furthermore, mutation-targeted treatment was significantly associated with a longer OS. *EGFR* mutation is the most common genetic alteration in East Asian lung adenocarcinoma patients [14,16]; hence, the survival benefits of mutation-targeted treatment in this group were mainly driven by the *EGFR* mutation and EGFR-TKI treatment. A recent Korean cohort study enrolled 8566 advanced lung adenocarcinoma patients to compare the effectiveness of NGS with that of single-gene testing. The results suggested no survival benefit of NGS [12]. Owing to the high prevalence of *EGFR* mutation in East Asia, both upfront NGS and single-gene testing may be reasonable [17]. However, genetic testing, per se, only provides information on driver mutation status. Patients’ outcome would improve once they have received the corresponding targeted therapy [5]. Apart from *EGFR* mutation, another 21.9% of Group 1 patients harbored targetable mutation(s), meaning that NGS may provide more opportunities of receiving mutation-targeted treatment for individuals harboring rare oncogenic drivers. Herein, our study did not conduct comparisons between upfront NGS and single-gene testing; rather, we emphasized the importance of mutation-targeted treatment.

Group 2 contained treatment-naïve patients with NGS performed after negative *EGFR*/*ALK*/*ROS1* results by conventional tests. Since most *EGFR*/*ALK*/*ROS1* mutations had been excluded, higher proportions of *HER2*, *MET*, *KRAS* G12C, and *BRAF* V600E mutations were detected. Mutation-targeted treatment was still associated with a favorable outcome among this population. However, the survival time was shorter than that of the other groups. As shown in Appendix A, the baseline characteristics were different among the various patient groups. There were more elders, males, smokers, and more patients with poor performance status in the Group 2 population. Because NGS was ordered after single-gene testing results, more time was needed to obtain the patients’ NGS data, which may have led to a decline in performance status, delays in starting treatment, and receiving the targeted therapy in the second-line setting or later [18]. According to previous studies, the worse efficacy of targeted therapies in the later lines of treatment could be anticipated [19,20,21]. Furthermore, treating patients with detectable driver mutations still appears to be worthwhile, because the subgroup analysis of driver mutation-positive patients revealed a significant difference in OS between individuals with and without the corresponding targeted treatment (Appendix A). Previous study suggested that females and never smokers with lung adenocarcinoma are more likely to harbor *EGFR/ALK/ROS1* mutation [22]. There are a total of 15 patients with *EGFR* or *ROS1* mutation in Group 2; of them, 73.3% were never smokers and 66.7% were females, showing the similar characteristics to previous reports. 

In the era of targeted therapy, successful treatment heavily depends on adequate tissue sampling and driver mutation detection [23]. Clinicians have to make every effort to obtain the targetable mutation information in order to improve patient outcome. When tumor tissue harbors scarce tumor cells, analysis of driver mutations, particularly the *EGFR* mutation, in cell-free DNA is recommended [24]. Group 3 contained post-treatment patients without known *EGFR*/*ALK*/*ROS1* mutation. All patients had a history of chemotherapy. There were 41.1% of patients with targetable mutation(s). The unavailability of tumor tissue, uncommon driver mutations, rare *EGFR* mutation subtypes, and the limitation of conventional genetic testing are possible reasons that may explain the missing detection of driver mutation(s) at diagnosis. Notably, as high as 84.4% of NGS testing in Group 3 was performed using liquid biopsy. Owing to advances in molecular technologies, liquid biopsy should be considered as a valid alternative for comprehensive genetic analysis [25]. A study by Ulivi et al. compared the concordance between Foundation One CDx and routine laboratory diagnostic assays, which disclosed perfect agreement regarding *EGFR* mutations (*n* = 60) [26]. However, thirty-one patients in Group 2 and Group 3 tested positive for *EGFR* mutations, including 11 with exon 20 insertions, 10 with L858R mutations, 5 with exon 19 deletions, and 5 with uncommon *EGFR* mutations. Previous studies have disclosed that PCR-based *EGFR* mutation assays may result in false negatives [27,28,29,30]. In the retrospective analysis of the EURTAC trial, false negative detection of both exon 19 deletions and L858R was observed [23]. In a retrospective cohort study, which included 62 NSCLC patients, the false negative rate of PCR methods was as high as 11.3% [29]. Heterogeneous subtypes of *EGFR* mutation, the various detection sensitivities of *EGFR* mutation assays, as well as the types and quality of specimens may influence the concordance between NGS and conventional diagnostic assays. Because of the heterogeneous conditions in the post-treatment population and limited patient numbers, numerically longer survival is only seen in patients receiving mutation-targeted treatment. Hence, it is necessary to develop a better algorithm for comprehensive genetic testing to avoid the missed detection of targetable mutations. 

Group 4 contained driver-positive patients experiencing disease progression on TKIs. Mutation-targeted treatment was associated with a numerically longer OS. The AURA3 study compared the efficacy of osimertinib versus platinum-pemetrexed for NSCLC patients with *EGFR* T790M and progression on first- or second-generation EGFR-TKI. Similarly, numerically longer OS was observed, but it was not statistically significant [31]. The resistance mechanisms of osimertinib are more heterogeneous, and the best treatment strategy remains uncertain [7]. Several early phase studies revealed the benefits of *MET* inhibitor(s). However, the results of long-term survival benefits are still pending [32,33,34]. A subgroup analysis of Group 4 patients also revealed a numerical longer survival time of individuals with positive T790M treated with osimertinib and those with MET amplification treated with MET inhibitor(s) (Appendix A). Although NGS has been widely used to explore the genetic characteristics of these resistant tumor cells, more studies are still needed to identify better treatment options.

Currently, not all of the targeted therapies are available in Taiwan, and some available medications are not reimbursed [35]. The differences between the percentage of targetable mutation(s) and the percentage of patients having received mutation-targeted treatment are a result of the availability of targeted therapies, reimbursement regulations, insurance coverage, and patients’ financial ability. To detect the mutation(s) only will not improve patients’ outcome. A shorter survival of individuals with targetable mutation(s) was observed universally among all groups of our patients if they had no opportunity to undergo corresponding targeted treatment (Appendix A). Physicians must try every effort to prescribe corresponding targeted treatment for patients with targetable mutation(s). 

This study has several limitations. First, it was a retrospective study conducted at a single center. However, we made efforts to ensure that we had a sufficient quantity and quality of data, including the patients’ characteristics, genetic alterations, and the outcome evaluation in order to analyze the prognostic impacts of NGS on different populations. Second, various NGS platforms were performed in the study. Although the coverage of genetic alterations was different, these platforms were all valid and covered most of the common targetable mutations of NSCLC and thus could represent real-world conditions in clinical practice. Third, NGS carries a higher cost, and the decision to conduct NGS testing may have been influenced by the patients’ socioeconomic status and the availability of treatments. These factors may have also affected the patients’ outcomes. Fourth, the patients’ characteristics and the distributions of targetable mutation(s) differ between East Asians and Caucasians [36]; hence, studies of different populations may yield various results. Fifth, different mutations may be associated with different outcomes. Subgroup analysis of each group considering a specific mutation will provide more information. However, it is limited by the patient numbers in the present study. 

## 4. Materials and Methods

### 4.1. Patients

This was a retrospective study, which included lung cancer patients diagnosed and treated at Taichung Veterans General Hospital from December 2018 to January 2024. To be eligible for the study, patients were required to have pathologically confirmed lung adenocarcinoma, inoperable stage III to IVB disease, and available NGS testing results as well as a precise history of diagnosis, treatment, and survival follow-up. With regard to NGS, results from both tumor tissue and liquid biopsy were acceptable for study inclusion. Patients were excluded if they had mixed histology other than adenocarcinoma, other active malignancy, stage III disease with an attempt of curative local therapy, or incomplete data records.

Patients were categorized into four groups according to their characteristics and timing of NGS testing, which included Group 1: treatment-naïve patients with upfront NGS testing, Group 2: treatment-naïve patients with NGS performed after negative epidermal growth factor receptor (*EGFR*)/anaplastic lymphoma kinase (*ALK*)/*ROS1* results by conventional tests, Group 3: treated patients without known *EGFR*/*ALK*/*ROS1* mutation, and Group 4: patients with known actionable driver mutation post-tyrosine kinase inhibitor (TKI) treatment.

### 4.2. Data Records for Analysis

Clinical data for analysis included patients’ age, gender, smoking status, Eastern Cooperative Oncology Group performance status (ECOG PS), histological type, specimen types for NGS testing, genetic alteration status, tumor stage, treatment history, and survival follow-up data. Lung cancer TNM (tumor, node, and metastases) staging was conducted according to the 8th edition of the American Joint Committee on Cancer (AJCC) staging system [37].

This study was approved by the Institutional Review Board of Taichung Veterans General Hospital (IRB No. CF12019 and CF20175). Written informed consent for clinical data records and genetic testing was obtained from all patients.

### 4.3. Next-Generation Sequencing Testing

All patients in this study underwent NGS testing using either tumor tissue or liquid biopsy. Commercialized NGS panels were applied for genetic testing (Appendix A). Tumor NGS panels included FoundationONE^®^ CDx (Foundation Medicine, Inc., Cambridge, MA, USA), Oncomine™ Focus Assay (Thermo Fisher Scientific, Waltham, MA, USA), ACTDrug^®^ + (ACT Genomics, Taipei, Taiwan), and ACTOnco^®^+ (ACT Genomics, Taipei, Taiwan). Liquid NGS panels included GUARDANT360^®^ (Guardant Health, Inc., Redwood City, CA, USA), FoundationONE^®^ LIQUID CDx (Foundation Medicine, Inc., Cambridge, MA, USA), ACTMonitor^®^ Lung (ACT Genomics, Taipei, Taiwan), TruSight™ Oncology 500 ctDNA (Illumina, Inc., San Diego, CA, USA), SOFIVA Cancer Monitor (SOFIVA Genomics Co., Ltd., New Taipei City, Taiwan), and ACT Cerebra™ (ACT Genomics, Taipei, Taiwan). For patients with both tumor tissue and liquid biopsy NGS data obtained simultaneously, we used the results of tumor NGS. For patients with more than one NGS test, we analyzed the first test results.

For Group 1, Group 2, and Group 3 patients, targetable mutations indicated *EGFR*, *ALK* fusion, *ROS1* fusion, *KRAS* G12C, *HER2*, *BRAF* V600E, *RET* fusion, *NTRK* fusion, *MET* exon 14 skipping, and *MET* amplification, according to the clinical practice guideline [3,4]. *EGFR* mutations occurring in non-exon 18–21 regions, non-G12C *KRAS* mutation, non-V600E *BRAF* mutation, *MET* point mutation, and non-fusion RET aberration were deemed to be untargetable mutations because of the uncertain efficacy of targeted therapies. Uncommon mutation types that were reported to be sensitive to and were being treated by a corresponding targeted therapy with clinical benefits were deemed to be targetable mutation(s). Patients without the aforementioned targetable and untargetable mutations were deemed to be undetectable. For Group 4 patients, the potential targetable mutations included *EGFR* T790M after progression on first- or second-generation EGFR-TKI, *EGFR* C797X after progression on osimertinib, *MET* amplification, and the emergence of other oncogenic driver(s). Mutation-targeted treatment was defined as medications corresponding to the actionable mutations.

### 4.4. Statistical Methods

Univariate analyses of the association between patient populations and baseline characteristics, the percentages of targetable mutations, as well as the percentages of mutation-targeted treatment were performed using Fisher’s exact test. The Kaplan–Meier method was used to estimate the survival time. Differences in survival time were analyzed by the log-rank test. The Cox proportional hazard model was used for multivariate analyses of the prognostic factors of survival outcomes. All statistical tests were carried out using SPSS 15.0 (SPSS Inc., Chicago, IL, USA). Two-tailed tests and *p* values < 0.05 for significance were implemented. Overall survival (OS) was the length of time from the diagnosis of advanced lung cancer to death from any cause, while post-NGS survival was defined as the length of time from NGS testing to death from any cause.

## 5. Conclusions

Regardless of NGS examination timing, a high proportion of targetable mutations can be identified, and mutation-targeted therapy was independently associated with a favorable outcome. The opportunities of receiving NGS-directed treatment and the survival benefits of mutation-targeted treatment varied among different patient populations. Treatment-naïve patients with upfront NGS testing could derive the greatest benefits.

## Figures and Tables

**Figure 1 ijms-25-06949-f001:**
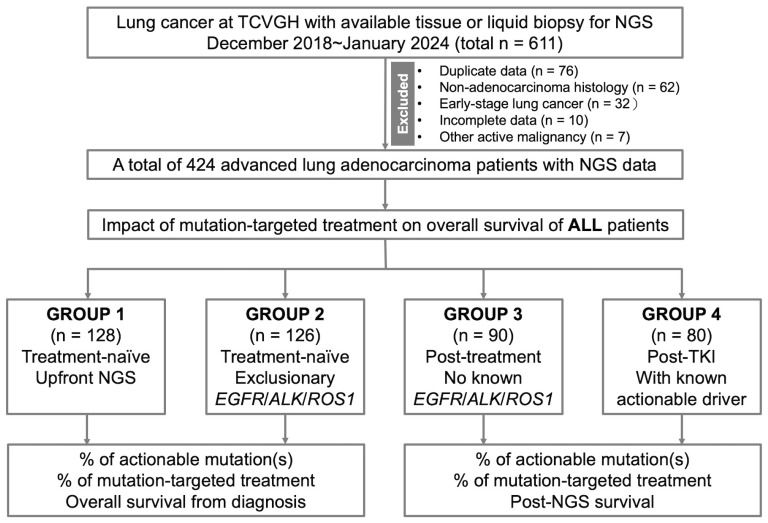
Patient selection, categorization, and outcome analysis.

**Figure 2 ijms-25-06949-f002:**
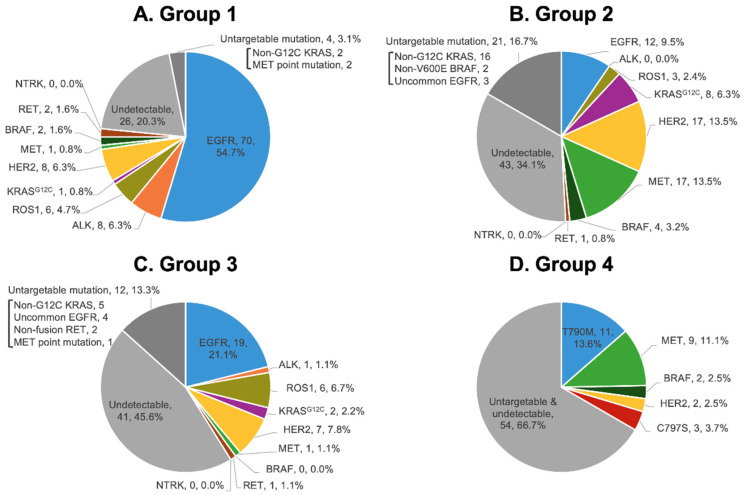
Distribution of targetable mutation(s) among various patient populations: Group 1 (**A**), Group 2 (**B**), Group 3 (**C**), and Group 4 (**D**).

**Figure 3 ijms-25-06949-f003:**
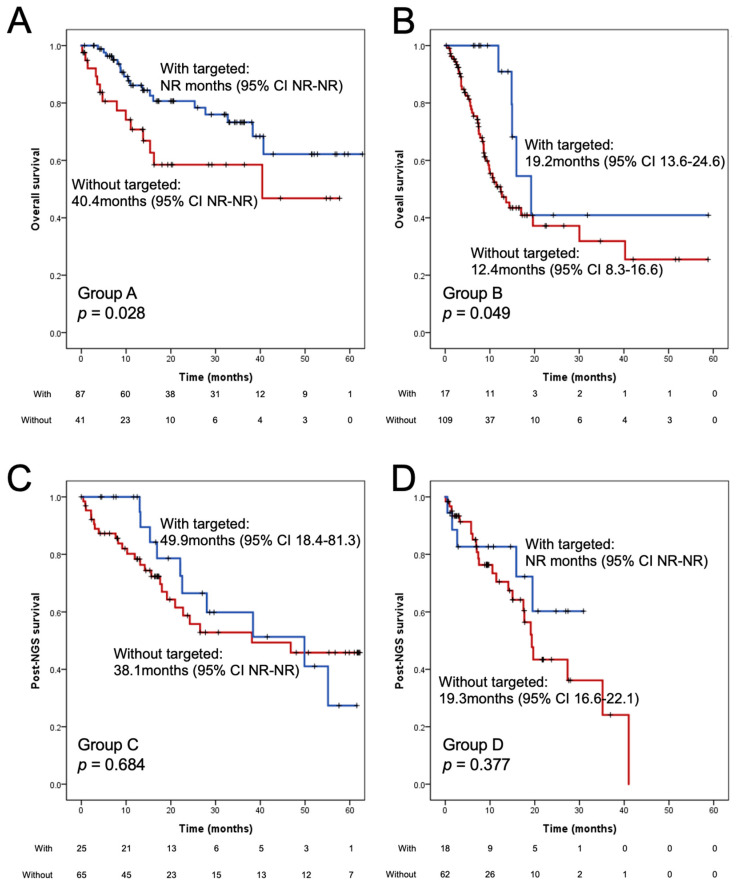
Survival benefits of next-generation sequencing and mutation-targeted treatment among various patient populations: Group 1 (**A**), Group 2 (**B**), Group 3 (**C**), and Group 4 (**D**).

**Figure 4 ijms-25-06949-f004:**
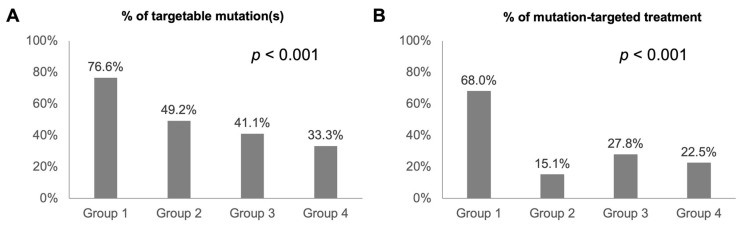
Percentage of targetable mutation(s) (**A**) and mutation-targeted treatment (**B**) among various patient populations.

**Table 1 ijms-25-06949-t001:** Demographic data and patients’ characteristics. ECOG PS, Eastern Cooperative Oncology Group performance status; NGS, next-generation sequencing. ^1^ Group 1: treatment-naïve, upfront NGS; Group 2: treatment-naïve, exclusionary *EGFR*/*ALK*/*ROS1*; Group 3: post-treatment, no known *EGFR*/*ALK*/*ROS1*; Group 4: known driver mutation and post-tyrosine kinase inhibitor treatment.

Characteristics	Total *n* = 424
Age, years, median (IQR)	61.6 (53.5–69.9)
Gender, *n* (%)	
Male	216 (50.9%)
Female	208 (49.1%)
Smoking status, *n* (%)	
Never smokers	241 (56.8%)
Smokers	183 (43.2%)
Tumor stage, *n* (%)	
Stage IIIB-C	16 (3.8%)
Stage IVA	161 (38.0%)
Stage IVB	247 (58.3%)
ECOG PS, *n* (%)	
0–1	371 (87.5%)
2 or more	53 (12.5%)
Specimens for NGS exam, *n* (%)	
Tumor tissue	211 (49.8%)
Liquid	213 (50.2%)
Patient categorization, *n* (%) ^1^	
Group 1	128 (30.2%)
Group 2	126 (29.7%)
Group 3	90 (21.2%)
Group 4	80 (18.9%)

**Table 2 ijms-25-06949-t002:** Various survival benefits among lung adenocarcinoma patients with next-generation sequencing examination. ^1^ Overall survival since diagnosis of advanced lung cancer for Group 1 and 2 and post-NGS survival for Group 3 and 4, respectively. ^2^ By log-rank test. ^3^ By Cox-proportion hazard model.

	Group 1	Group 2	Group 3	Group 4
Patient No.	128	126	90	80
Survival, mo ^1^				
Targeted	NR (NR-NR)	19.2 (13.9–24.6)	49.9 (18.4–81.3)	NR (NR-NR)
Not targeted	40.4 (NR-NR)	12.4 (8.3–16.6)	38.1 (NR-NR)	19.3 (16.6–22.1)
*p* value ^2^	0.028	0.049	0.684	0.377
Hazard ratio (95% CI)	0.46 (0.23–0.93)	0.41 (0.16–1.03)	0.86 (0.41–1.79)	0.38 (0.24–1.72)
*p* value ^3^	0.032	0.057	0.684	0.381

## Data Availability

The data presented in this study are available on request from the corresponding author.

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
