# Peer review of "Benefits of NGS in Advanced Lung Adenocarcinoma Vary by Populations and Timing of Examination"

_ijms, 2024, doi:10.3390/ijms25136949_

Round 1

Reviewer 1 Report

Comments and Suggestions for Authors

I read with interest the manuscript by Po-Hsin Lee and colleagues, in which a retrospective real-life analysis of a large cohort advanced Non-Small Cell Lung Cancer patients is performed from a molecular testing perspective for patients clinical benefit. The authors question the utility of NGS analysis and its timing in different patients sub-cohorts, identified by treatment and molecular testing. Several studies have been conducted to assess the utility of NGS platforms, even from an economic, time-consuming and clinical utility point of view. The article is interesting even though lacking of novelty, and some points need to be clarified in terms of analysis.

1-      Patients composing group 2 were tested for activating mutations with “conventional tests” but they were wild-type for investigated alterations. Moreover, “more elders, males, smokers”, i.e. the clinical features less associated to a molecular targetable alteration (PMID 35081265). On the other hand, by NGS analysis, >10% of patients were diagnosed with an activating mutation in EGFR/ALK/ROS1. Deeper discussion related to this point would be welcome, even considering the comparison of NGS platforms with “conventional tests” (PMID 35625958).

2-      Group 3 is composed by “treated patients without known EGFR/ALK/ROS1 mutation”; this is an interesting group of the study, as it further underlines the need of molecular testing for advanced NSCLC; along the Discussion, limitations in NGS testing are provided, authors could deepen this, being the main focus of the article, underlining the role of liquid biopsy, sampling (PMID 31857950; 28101783). Moreover, I am not sure about the added value of group 4 in this study.

3-      The authors show that a considerable part of gene alterations are “undetectable”; do they mean undetectable “targetable” mutations? Please better describe, even explaining why in the group 4 the untargetable and undetectable are considered together.

4-      Retrospective observational window is “December 2018 to January 2024”, which makes controversial analysis for patients survival, as part of patients had a short follow-up period. Tables with number at risk for each graph of figure 3 would be welcome.

5-      Graph A of figure 3 is not so informative: it is well known that NSCLC patients treated with TKIs experience a prolonged OS, and this is not related to NGS testing, as highlighted in the figure legend.

6-      Supplementary data are poorly described and they add nothing more to the text, please address them or remove.

Comments on the Quality of English Language

Minor check 

Author Response

Dear Editor-in-Chief and reviewers:

    We appreciate the editor’s and reviewers’ recommendations and encouragement. The followings are the responses and major changes in the revised manuscript. We do our best to revise this manuscript and answer the reviewers’ questions. We appreciate your kindness for reconsideration of our paper, possibly published in International Journal of Molecular Sciences. 

Submission ID: ijms-3050485

Reviewer 1:

  1. I read with interest the manuscript by Po-Hsin Lee and colleagues, in which a retrospective real-life analysis of a large cohort advanced Non-Small Cell Lung Cancer patients is performed from a molecular testing perspective for patients’ clinical benefit. The authors question the utility of NGS analysis and its timing in different patients’ sub-cohorts, identified by treatment and molecular testing. Several studies have been conducted to assess the utility of NGS platforms, even from an economic, time-consuming and clinical utility point of view. The article is interesting even though lacking of novelty, and some points need to be clarified in terms of analysis.

Response: We appreciate the reviewer’s comments and encouragement. We’ll try to revise all the points recommended by reviewers.

  1. Patients composing group 2 were tested for activating mutations with “conventional tests” but they were wild-type for investigated alterations. Moreover, “more elders, males, smokers”, i.e. the clinical features less associated to a molecular targetable alteration (PMID 35081265). On the other hand, by NGS analysis, >10% of patients were diagnosed with an activating mutation in EGFR/ALK/ROS1. Deeper discussion related to this point would be welcome, even considering the comparison of NGS platforms with “conventional tests” (PMID 35625958).

Response: We appreciate the reviewer’s recommendations. Group 2 contained more elders, males, and smokers, which are the characteristics of whole group. Of them, 15 patients harbored EGFR or ROS1 mutation; 73.3% and 66.7% of them were never smokers and females, respectively. We have added statements for better description and cited corresponding reference (page 8, line 247-251). We’ve also enhanced the description regarding the concordance between NGS and conventional detection methods. Heterogeneous subtypes of EGFR mutation, various detection sensitivity of EGFR mutation assays, as well as the types and quality of specimens may influence the concordance between NGS and conventional diagnostic assays. Corresponding reference was also cited (page 8, line 264-266 and line 273-275).

  1. Group 3 is composed by “treated patients without known EGFR/ALK/ROS1 mutation”; this is an interesting group of the study, as it further underlines the need of molecular testing for advanced NSCLC; along the Discussion, limitations in NGS testing are provided, authors could deepen this, being the main focus of the article, underlining the role of liquid biopsy, sampling (PMID 31857950; 28101783). Moreover, I am not sure about the added value of group 4 in this study.

Response: We appreciate the reviewer’s recommendations and agree with reviewer’s points. We enhanced the statement regarding the need of molecular testing for advanced NSCLC and highlighted the role of liquid biopsy in the Discussion of Group 3 patients (page 8, line 252-256) with citing corresponding references. Group 4 contained TKI-treated patients, which represents an important population requiring NGS testing. In clinical practice and research, physicians tried many efforts to perform re-biopsy and to explore the resistance mechanism using NGS testing. However, no clinical trials can well demonstrate the overall survival benefit of this kind approach. Herein, we disclosed the real-world condition, revealed the trends of longer survival of mutation-targeted therapy, and highlighted the unmet need of these patients. As we mentioned, results of many studies are still pending, and more studies are still needed to identify better treatment options (page 9, paragraph line 280-294).

  1. The authors show that a considerable part of gene alterations are “undetectable”; do they mean undetectable “targetable” mutations? Please better describe, even explaining why in the group 4 the untargetable and undetectable are considered together.

Response: We appreciate the reviewer’s comments. We added a sentence “Patients without aforementioned targetable and untargetable mutations were deemed to be undetectable.” in the section of “Methods” for better description (page 10, line 364-365). As we mentioned in the same section, for Group 4 patients (page 10, line 365-368), potential targetable mutations included EGFR T790M after progression on first- or second-generation EGFR-TKI, EGFR C797X after progression on osimertinib, MET amplification, and the emergence of other oncogenic driver(s). Other resistance mechanisms, such as PIK3CA and cell cycle gene alterations, have been reported and could be detected by NGS. However, their clinical significance remains unknown and there is no suitable treatment options. Moreover, the patient number of each genetic alteration is low; hence, we categorized these genetic alterations (untargetable) and those with undetectable together.

  1. Retrospective observational window is “December 2018 to January 2024”, which makes controversial analysis for patients survival, as part of patients had a short follow-up period. Tables with number at risk for each graph of figure 3 would be welcome.

Response: We appreciate the reviewer’s recommendations. We have added the number at risk for each graph of Figure 3 in the revised file (page 5).

  1. Graph A of figure 3 is not so informative: it is well known that NSCLC patients treated with TKIs experience a prolonged OS, and this is not related to NGS testing, as highlighted in the figure legend.

Response: We appreciate the reviewer’s comments. We revised the Figure 3 and moved original Figure 3A to supplementary file as a Supplementary Figure (Figure S1).

  1. Supplementary data are poorly described and they add nothing more to the text, please address them or remove.

Response: We appreciate the reviewer’s comments. In order to respond to another reviewer’s comments, we reserved the supplementary figures. We revised the original Figure S2 and merged them as Figure S2 and address the importance as “To detect the mutation(s) only will not improve patients’ outcome. A shorter survival of individuals with targetable mutation(s) was observed universally among all groups of our patients if they had no opportunity to undergo corresponding targeted treatment (Figure S2 and S3). Physicians must try every effort to prescribe corresponding targeted treatment for patients with targetable mutation(s).” in the section of “Discussion” (page 9, line 297-301). We also reserved the Figure S3 because it represented the outcome of different subgroups of Group 4 patients and we enhanced the descriptions in page 9, line 288-290.

The changes made in revised manuscript are marked by Red Font.

Correspondence to: Jeng-Sen Tseng

Jeng-Sen Tseng, M.D., Ph.D.

Division of Chest Medicine, Department of Internal Medicine, Taichung Veterans General Hospital, No.1650, Sect. 4, Taiwan Blvd., Taichung, 407 Taiwan

E-mail: tzeng64@gmail.com

TEL: +886-4-23592525, ext. 3232

FAX: +886-4-23741320

Reviewer 2 Report

Comments and Suggestions for Authors

This manuscript presents research that falls within the trend towards personalized medicine. The authors show results of next-generation sequencing (NGS) in advanced lung adenocarcinoma and its impact on survival. The patients were divided into four groups: Group1: treatment naïve, upfront NGS; Group2: Treatment-naïve, exclusionary EGFR/ALK/ROS1; Group3: post-treatment, no known EGFR/ALK/ROS1; Group4: known driver mutation and post-TKI treatment. As summary, the authors claim that Mutation-targeted therapy is associated with a favorable outcome  and Treatment-naïve patients (Group1) may derive the greatest benefit. These conclusions do not bring anything new to science. I think that the authors do not used the potential of the NGS data  and they would get more interesting results with other classification than mentioned 4 Groups. Each group should be divided into subgroups with taking into account a specific mutation, e.g. EGFR mutation, and comparing the results in: not treatment, post-treatment, etc. I therefore recommend to reject the manuscript in this form.

Author Response

Dear Editor-in-Chief and reviewers:

    We appreciate the editor’s and reviewers’ recommendations and encouragement. The followings are the responses and major changes in the revised manuscript. We do our best to revise this manuscript and answer the reviewers’ questions. We appreciate your kindness for reconsideration of our paper, possibly published in International Journal of Molecular Sciences. 

Submission ID: ijms-3050485

Reviewer 2:

  1. This manuscript presents research that falls within the trend towards personalized medicine. The authors show results of next-generation sequencing (NGS) in advanced lung adenocarcinoma and its impact on survival. The patients were divided into four groups: Group1: treatment naïve, upfront NGS; Group2: Treatment-naïve, exclusionary EGFR/ALK/ROS1; Group3: post-treatment, no known EGFR/ALK/ROS1; Group4: known driver mutation and post-TKI treatment. As summary, the authors claim that Mutation-targeted therapy is associated with a favorable outcome and Treatment-naïve patients (Group1) may derive the greatest benefit. These conclusions do not bring anything new to science. I think that the authors do not used the potential of the NGS data and they would get more interesting results with other classification than mentioned 4 Groups. Each group should be divided into subgroups with taking into account a specific mutation, e.g. EGFR mutation, and comparing the results in: not treatment, post-treatment, etc. I therefore recommend to reject the manuscript in this form.

Response: We appreciate the reviewer’s comments. We divided our patients into four groups, which represent the four most common and practical situations that clinicians will order NGS in clinical practice. We agree with reviewer’s points that it is interesting to further evaluate the specific mutation among each group. However, the analysis is limited by patient numbers. We added this point to the section of limitation (page 9, 314-216). We further performed subgroup analysis of each group with considering “targeted treatment” and “no targeted treatment” of individuals harboring targetable mutation(s). The results are shown in Figure S2 and S3. We added “To detect the mutation(s) only will not improve patients’ outcome. A shorter survival of individuals with targetable mutation(s) was observed universally among all groups of our patients if they had no opportunity to undergo corresponding targeted treatment (Figure S2 and S3). Physicians must try every effort to prescribe corresponding targeted treatment for patients with targetable mutation(s).” to address the importance of corresponding targeted treatment in the section of “Discussion” (page 9, line 297-301). As we mentioned, though NGS has been widely applied in the management of advanced lung adenocarcinoma, its impact on survival and the optimal timing for the examination remain uncertain. Herein, we try to demonstrate the different opportunities of receiving NGS-directed treatment and the various survival benefits of mutation-targeted treatment among different patient populations. There are many interesting and important points of NGS data. The analyses of these data (e.g., the co-mutation status) and its impacts on patient outcomes are undertaken in another studies.

The changes made in revised manuscript are marked by Red Font.

Correspondence to: Jeng-Sen Tseng

Jeng-Sen Tseng, M.D., Ph.D.

Division of Chest Medicine, Department of Internal Medicine, Taichung Veterans General Hospital, No.1650, Sect. 4, Taiwan Blvd., Taichung, 407 Taiwan

E-mail: tzeng64@gmail.com

TEL: +886-4-23592525, ext. 3232

FAX: +886-4-23741320

Round 2

Reviewer 1 Report

Comments and Suggestions for Authors

The authors improved the manuscript and provided convincing replies to the raised concerns.

This reviewer suggest that the manuscript is now ready for publication.

Comments on the Quality of English Language

Minor spell check

Reviewer 2 Report

Comments and Suggestions for Authors

I accept the authors' explanations and corrections.